# MVGamba: Unify 3D Content Generation as State Space Sequence Modeling

**Xuanyu Yi**[1,5*]  **Zike Wu**[3*]  **Qiuhong Shen**[2*]  **Qingshan Xu**[1]  **Pan Zhou**[4]
**Joo-Hwee Lim**[5]  **Shuicheng Yan**[6]  **Xinchao Wang**[2]  **Hanwang Zhang**[1]

[1]Nanyang Technological University  [2]National University of Singapore
[3]University of British Columbia  [4]Singapore Management University
[5]Institute for Infocomm Research  [6]Skywork AI

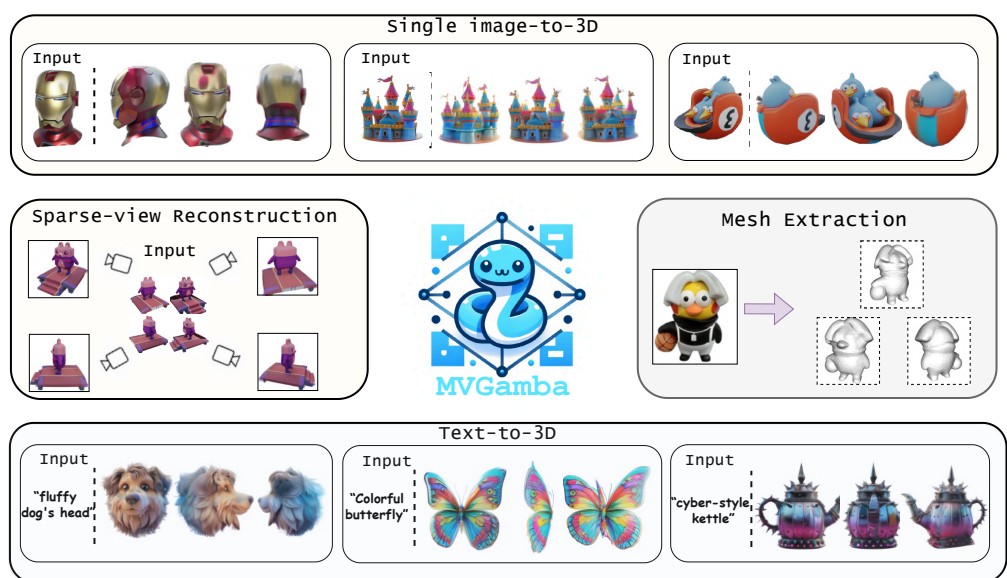

Figure 1: MVGamba is a unified 3D generation framework build on Gaussian Splatting, which can generate high-quality 3D contents in a feed-forward manner in sub-seconds.

## Abstract

Recent 3D large reconstruction models (LRMs) can generate high-quality 3D content in sub-seconds by integrating multi-view diffusion models with scalable multi-view reconstructors. Current works further leverage 3D Gaussian Splatting as 3D representation for improved visual quality and rendering efficiency. However, we observe that existing Gaussian reconstruction models often suffer from multi-view inconsistency and blurred textures. We attribute this to the compromise of multi-view information propagation in favor of adopting powerful yet computationally intensive architectures (*e.g.*, Transformers). To address this issue, we introduce MVGamba, a general and lightweight Gaussian reconstruction model featuring a multi-view Gaussian reconstructor based on the RNN-like State Space Model (SSM). Our Gaussian reconstructor propagates causal context containing multi-view information for cross-view self-refinement while generating a long

---

[*]Equal Contribution

38th Conference on Neural Information Processing Systems (NeurIPS 2024).

sequence of Gaussians for fine-detail modeling with linear complexity. With off-the-shelf multi-view diffusion models integrated, MVGamba unifies 3D generation tasks from a single image, sparse images, or text prompts. Extensive experiments demonstrate that MVGamba outperforms state-of-the-art baselines in all 3D content generation scenarios with approximately only $0.1\times$ of the model size. The codes are available at https://github.com/SkyworkAI/MVGamba.

# 1  Introduction

We address the challenge of crafting 3D content from a single image, sparse-view images, or text input, which can facilitate a broad range of applications, *e.g.*, Virtual Reality, immersive filming, digital gaming and animation. Previous research on 3D generation has investigated distilling 2D diffusion priors into 3D representations via score distillation sampling (SDS) [1]. Although these optimization-based approaches exhibit strong zero-shot generation capability with high-fidelity rendering quality [2–5], they are extremely time- and memory-intensive, often requiring hours to produce a single 3D asset, thus not practical for a real-world scenario.

With the advent of large-scale open-world 3D datasets [6–8], recent 3D large reconstruction models (LRMs) [9–12] integrate multi-view diffusion models [13–15] with scalable multi-view 3D reconstructor to regress a certain 3D representation (*e.g.* Triplane-NeRF [16, 17], mesh) in a feed-forward manner. Specifically, current LRMs [18–20] adopt a *one image (or text) → mulit-view images → 3D* diagram to predict 3D Gaussian Splatting (3DGS) [21] parameters, thereby ensuring the rendering efficiency while preserving fine details. Given a single image or text prompt, they first generate a set of images using multi-view diffusion models, which are then fed into a multi-view reconstructor (*e.g.*, U-Net [22] or Transformer [23]), mapping image tokens to 3D Gaussians with superior generation speed and unprecedented quality.

However, we observe that existing feed-forward Gaussian reconstruction models typically adopt powerful yet computationally intensive architectures [23, 24] to generate long sequences of Gaussians for intricate 3D modeling. Such approaches inevitably compromise the integrity of multi-view information propagation to manage computational costs. For instance, they use local [18] or mixed [19] attention on limited multi-view image tokens or even deal each view separately and simply merge the predicted Gaussians afterwards [20]. Consequently, the generated 3D models often suffer from multi-view inconsistency and blurred textures, as illustrated in Figure 2(a). These issues indicate that current compromise strategies fail to translate into coherent, high-quality outputs in practice. This raises a crucial question: *How can we preserve the integrity of multi-view information while efficiently generating a sufficiently long sequence of Gaussians?*

To address this issue, in this paper, we introduce **M**ulti-**V**iew **G**aussian **M**amba (MVGamba), a general and lightweight Gaussian reconstruction model. At its core, MVGamba features a multi-view Gaussian reconstructor based on the recently introduced RNN-like architecture Mamba [25], which expands the given multi-view images into a long sequence of 3D Gaussian tokens and processes them recurrently in a causal manner. By adopting causal context propagation, our approach efficiently maintains multi-view information integrity and further enables cross-view self-refinement from earlier to current views. Additionally, our Gaussian reconstructor enables the fine-detailed generation of long Gaussian sequences with linear complexity [26, 27] in a single forward process, eliminating the need for any post hoc operations used in previous work.

More concretely, we first patchify the multi-view images into $N$ tokens and rearrange them according to a cross-scan order [27, 28], resulting in $4 \times N$ image tokens for selective scanning. These tokens are then processed through a series of Mamba blocks for state space sequence modeling. Subsequently, we feed the output Gaussian sequence into a lightweight Multi-Layer Perceptron (MLP) for channel-wise knowledge selection, followed by a set of linear decoders to obtain the Gaussian parameters representing high-quality 3D content (Sec. 3.2). Compared to previous LRMs [11, 29, 30], our MVGamba features many computationally efficient components: a single-layer 2D convolution image tokenizer replaces the pre-trained DINO [31] transformer encoder, a lightweight MLP combined with linear decoders replaces the deep MLP decoder, and most importantly, linear complexity Mamba blocks replace quadratic complexity Transformer blocks (Figure 2(b)). Together, these designs ensure efficient training and inference while achieving higher generation quality (Sec. 4). Moreover, to

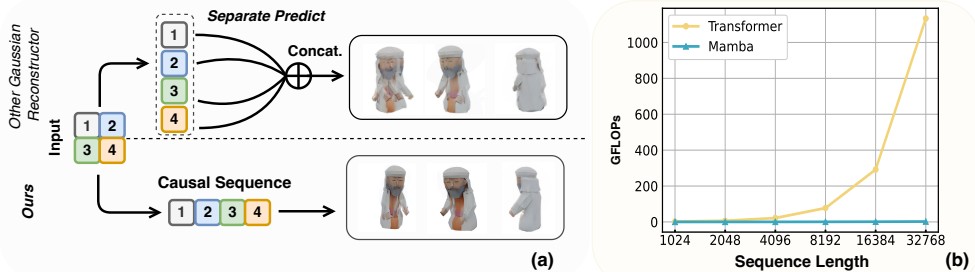

Figure 2: **(a)** Previous Gaussian reconstruction models sacrifice the integrity of multi-view information for computationally intensive architectures, resulting in multi-view inconsistency and blurred textures. **(b)** Comparison of FLOPs between self-attention in Transformers and SSM in Mamba. Detailed FLOPs data are provided in Table 3.

directly convert the generated Gaussians into smooth textured polygonal meshes, we alternatively incorporate a 3DGS variant — 2DGS [32] — for accurate geometric modeling and mesh extraction.

We conducted comprehensive qualitative and quantitative experiments to verify the efficacy of our proposed MVGamba. The experimental results demonstrate that MVGamba (49M parameters) outperforms other latest LRMs [19, 29, 33] and even optimization-based methods [34, 35] on the task of text-to-3D generation, single-view reconstruction and sparse-view reconstruction with roughly only $0.1\times$ of the model size. The contributions and novelties of our paper are summarized as follows:

- We point out that directly generating a sufficiently long sequence of Gaussians with full multi-view information is crucial for consistent and fine-detailed 3D generation.
- We introduce MVGamba, a novel feed-forward pipeline that incorporates causal context propagation for cross-view self-refinement, allowing the efficient generation of long sequences of 2D/3D Gaussians for high-quality 3D content modeling.
- Extensive experiments demonstrate that MVGamba is a potentially *general* solution for 3D content generation, including text-to-3D, image-to-3D and sparse-view reconstruction task.

## 2 Related Work

**3D Generation.** Previous approaches for generating high-fidelity 3D models predominantly used SDS-based optimization techniques [1, 36] and their variants [2–5, 34, 37]. These methods yield high-quality 3D generations but require hours for the per-instance optimization process to converge. Pioneered by the large reconstruction model (LRM) [29], recent works [10, 11, 30, 38] show that image tokens can be directly mapped to 3D representations, typically triplane-NeRF, in a feed-forward manner via a scalable transformer-based architecture [23] with large-scale 3D training data [6–8]. Among them, Instant3D [11] integrates LRM with multi-view image diffusion models [13–15, 39, 40], using four generated images for better quality. To avoid inefficient volume rendering and limited triplane resolution, some concurrent works [18–20] follow Instant3D and introduce 3D Gaussian Splatting [21] into sparse-view LRM variants. Specifically, GRM [18] and GS-LRM [20] use pixel-aligned Gaussian with a pure transformer-based reconstruction model, increasing the number of Gaussians through image feature upsampling and per-pixel merge operations. LGM [19] combines the 3D Gaussians from different views using a convolution-based asymmetric U-Net [22]. Our MVGamba, on the other hand, directly processes multi-view conditions causally, recurrently generating a long sequence of Gaussians for coherent and high-fidelity 3D modeling.

**Mamba model for visual applications.** Recent advancements in State Space Models (SSMs) [17, 41, 42], notably Mamba [25], have gained prominence in long sequence modeling for harmonizing computational efficiency and model versatility [43–46]. Following Mamba's progress, there has been a surge in applying this framework to critical vision domains, including generic vision backbones [26, 27, 47, 48], multi-modal streams [49, 50], and vertical applications, especially in medical image processing [51–56]. Specifically, VMamba [27] pioneers a purely Mamba-based backbone to handle intensive prediction tasks. Similarly, Vim [26] leverages bidirectional SSMs for data-dependent global visual context without image-specific biases. Subsequent works progress with advanced selective scanning algorithms [47, 48], integration with other networks [57, 58], and

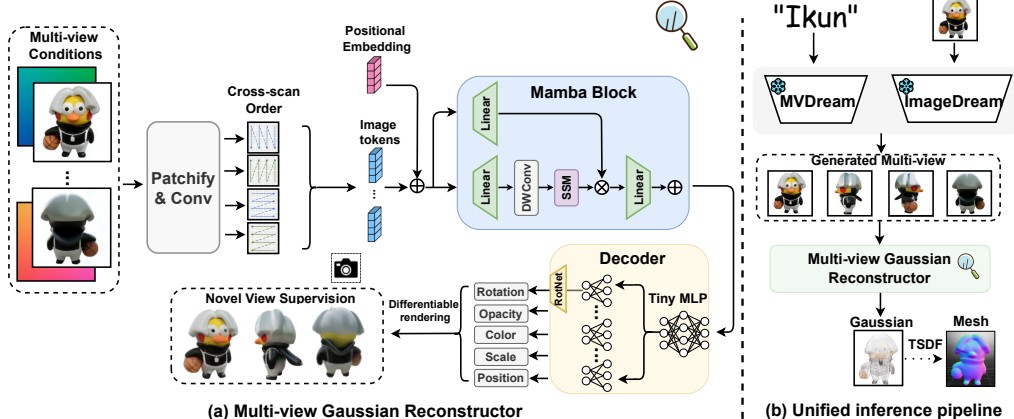

**(a) Multi-view Gaussian Reconstructor**        **(b) Unified inference pipeline**

Figure 3: **(a) Multi-view Gaussian reconstructor (Sec. 3.2):** Multi-view inputs with ray embedding are used for causal sequence modeling, predicting Gaussians rendered at novel views and supervised with ground truth images. **(b) Unified inference pipeline (Sec. 3.4):** MVGamba combines multi-view diffusion models and Gaussian reconstructor to generate high-quality 3D content in sub-seconds.

adapted structural designs [59, 59, 60]. Concurrently, Gamba [61] marries Mamba with 3DGS for single-view reconstruction with limited texture quality and generalization capacity. In this paper, we explore and demonstrate the efficiency and long-sequence modeling capacity of Mamba in various 3D generation tasks with large-scale pre-training.

## 3 Method

In this section, we present our MVGamba, designed to efficiently generate 3D content through a two-stage pipeline. In the first stage, we utilize off-the-shelf multi-view diffusion models, including MVDream [14] and ImageDream [13], to generate multi-view images based on an input text prompt or a single image. In the second stage, equipped with this multi-view image generator, we introduce an SSM-based multi-view reconstructor to generate Gaussians from multi-view images. Specifically, we first provide a brief overview of 3D Gaussian splatting and its variants (Sec. 3.1). Next, we describe the core architecture of our multi-view Gaussian reconstructor (Sec. 3.2), followed by detailed elaboration of our robust training objectives (Sec. 3.3). With above large-scale pre-training, we are able to chain these two stages to produce high-fidelity 3D content in seconds (Sec. 3.4).

### 3.1 Preliminary: Gaussian Splatting

Introduced by [21], vanilla 3D Gaussian Splatting (3DGS) fits a 3D scene from multi-view images with a collection of 3D Gaussians. Variants of Gaussian Splatting [62, 63], typically 2DGS, leverage 2D Gaussian primitives instead, excelling in the vanilla version for more accurate geometry reconstruction. Generally, each Gaussian is composed of its 3D center $\mu \in \mathbb{R}^3$, 3D scale $s \in \mathbb{R}^3$ or 2D scale * $s \in \mathbb{R}^2$, associated color $c \in \mathbb{R}^3$, opacity $\alpha \in \mathbb{R}$, and a rotation quaternion $q \in \mathbb{R}^4$. These parameters can be collectively denoted by $\mathcal{G}$, with $\mathcal{G}_i = \{\mu_i, s_i, c_i, \alpha_i, q_i\}$ denoting the parameter of the $i$-th Gaussian. These Gaussians can then be splatted onto the image plane and rendered in real time via the differentiable tiled rasterizer [21, 62].

### 3.2 SSM-based Gaussian reconstructor

The core of MVGamba is a feed-forward multi-view Gaussian reconstructor. As depicted in Figure 3(a), our reconstructor transforms multi-view input images with camera embedding [64] into 3D contents represented by 3D Gaussians [21] or its variants [62, 63] in a feed-forward manner. This reconstructor comprises an SSM-based processor to expand and process multi-view image tokens as Gaussian sequences, and a light-weight Gaussian decoder to predict attributes for each Gaussian.

---

*2DGS collapses the 3D volume into a set of 2D oriented planar Gaussian disks to model surfaces intrinsically.

**Expanding multi-view images as sequences.** Given a posed multi-view image set $\{v_i, \pi_i\}$, we first densely embed the camera pose $\pi_i \in \mathbb{R}^{4\times4}$ for each view $v_i \in \mathbb{R}^{H\times W\times3}$ using Plücker rays [30], denoted as $\mathcal{P}_i \in \mathbb{R}^{H\times W\times6}$. The pixel values and ray embeddings are concatenated into a 9-channel fused map, which is then tokenized using a non-overlapping convolution with a kernel size of $p \times p$:

$$\mathbf{V}_i = \text{Conv}(\text{Concat}(v_i, \mathcal{P}_i)), \tag{1}$$

where $\mathbf{V}_i \in \mathbb{R}^{h\times w\times C}$ is the tokenized feature map; $h = H/p$, $w = W/p$; $C$ is the embedding dimension. Note that considering the light-weight architectural design, our image tokenizer is much simpler than the pre-trained DINO [31] utilized by previous LRMs, which we empirically find to be redundant for low-level 3D reconstruction. With the tokenized multi-view features, we then adopt a cross-scan order [27] to rearrange them as sequence. Specifically, we scan the image tokens sequentially along four different directions: `top-left` $\rightarrow$ `bottom-right`, `bottom-right` $\rightarrow$ `top-left`, `top-right` $\rightarrow$ `bottom-left`, and `bottom-left` $\rightarrow$ `top-right`, which allows each token to integrate information from all adjacent tokens. This cross-scan rearrangement results in a sequence that is $4\times$ longer:

$$\mathbf{X} = \text{P}_{\text{scan}}(\text{Concat}(\{\mathbf{V}_i\})), \tag{2}$$

where $\mathbf{X} \in \mathbb{R}^{4Nhw\times C}$ denotes the expanded Gaussian sequences, $N$ denotes the view number, and $\text{P}_{\text{scan}}$ denotes the cross-scan operation on each view.

**Causal sequence modeling with State Space Model.** Inspired by [25, 61], we model the Gaussian sequences via an adapted SSM-based processor. In detail, given the expanded Gaussian sequence $\mathbf{X}$, we first add a learnable positional embedding $\mathbf{E}$ element-wise to it and derive the initial Gaussian sequence $\mathbf{X}_0$. Then, we feed $\mathbf{X}_0$ into $L$ stacked SSM layers for recurrent causal sequence modeling, formulated as:

$$\mathbf{X}_k = \text{SSM}_k(\mathbf{X}_{k-1}; A_k, B_k, \Delta_k) \tag{3}$$

where $\mathbf{X}_k$ denotes Gaussian sequence output by the $k$-th layer, and $\text{SSM}_k$ denotes the $k$-th SSM layer with vanilla Mamba [25] structure; $A_k$, $B_k$ and $\Delta_k$ are parameters of SSM layer dependent on the input sequence $\mathbf{X}_{k-1}$. Note that we are modeling Gaussian sequences rather than integrating spatial information as in existing vision Mamba models [26, 27, 47]. Therefore, we adopt 1D convolution instead of 2D convolution in the Mamba blocks, similar to other sequential modeling SSMs [25, 58, 65]. Through state space sequence modeling, we successfully propagate the causal context containing the multi-view information from earlier states to later states with linear complexity. This approach efficiently incorporates multi-view information causally from the initial condition onward, thereby making full use of all Gaussian tokens through cross-view self-refinement. As discussed in Sec. 5, this causal sequential generation of Gaussian tokens provides the model with unprecedented robustness and self-correction abilities, even under inconsistent or noisy input conditions.

**Decoding causal token sequences into Gaussians.** Each token in the processed causal sequence $\mathbf{X}_L$ is treated as a separate 3D Gaussian token. We first apply a single hidden layer MLP to $\mathbf{X}_L$, where the width of the hidden layer is $4C$ and the output channels revert to $C$. This process is denoted as $\mathbf{Z} = \mathbf{MLP}(\mathbf{X}_L)$, which aims for channel-wise knowledge selection [24, 66, 67]. We then apply sub-heads to derive each attribute of 3DGS with separate linear projections. Specifically, we predict the position [61] by discretizing the coordinates where position $\mu_i$ is clamped to $[-1, 1]^3$. The scale $s_i$ is predicted with a learnable linear projection followed by a $\mathrm{softplus}$ activation. The opacity $\alpha_i$ is predicted with a linear projection followed by $\mathrm{sigmoid}$ activation. Regarding the color attribute $c_i$, we predict the RGB values instead of the spherical harmonics adopted by the original 3DGS[21], as our reconstructor is mainly trained on synthetic 3D datasets free of light variation.

However, unlike other Gaussian attributes, the rotation quaternions are quite sensitive and difficult to predict directly, and hence are often set canonical isotropic and fixed in several recent works [61, 68]. On the other hand, some works [19, 33] predict the rotation without any constraints, but this often causes artifacts and corrupted generations in practice. To address this issue, we design a novel rotation decoder, dubbed RotNet, which balances prediction flexibility and restriction. Our RotNet consists of a set of 32 pre-defined rotation quaternions, denoted as $\mathbf{T}$, which forms a canonical rotation space, and a learnable linear projection matrix $\Theta$ to predict the logits of these quaternions. The predicted logits are then transformed into a probability distribution using the Gumbel-Softmax [69], enabling differentiation through the discrete selection process by adding noise sampled from the Gumbel distribution to the logits $\mathbf{p} \in \mathbb{R}^{32}$ before applying the softmax function:

$$\mathbf{p} = \text{softmax}(\Theta\mathbf{Z} + \mathbf{g}), \quad \text{where} \quad g_k = -\log(-\log(u_k)), \quad u_k \sim \text{Uniform}(0, 1). \tag{4}$$

In this way, we convert the rotation prediction into a 32-class classification task in a fully differentiable way, which allows for direct selection via the argmax operation during inference. We refer to *Appendix D* for more detailed explanations. These decoded Gaussians are finally passed into the differentiable rasterization pipeline [21] for image-level supervision.

### 3.3 Stable Training of MVGamba

**Bridging the training-inference gap.** In the training phase, multi-view images are collected from the ground-truth blendering of 3D objects, while they are generated by diffusion models during inference. To mitigate such domain gap: (1) Following LGM [19], we leverage the grid distortion and orbital camera jitter as two data augmentations with a 30% probability to simulate inconsistent pixels and inaccurate camera poses, respectively. (2) We directly use ImageDream [13] as a synthetic data engine to generate multi-view images input and conduct a joint training with the ground-truth renderings from the 3D training dataset. In practice, with a 5% chance, we train MVGamba with synthetic input to mimic the inference pattern for more robust generation results.

**Overall training objective.** During the training phase, we differentiably render the RGB image $v_i$ and alpha mask $v_i^\alpha$ of the $N = 4$ input views and another six novel views for image-level supervision. Our final objective then comprises four key terms:

$$\mathcal{L} = \sum_{v_i} \frac{1}{||v_i||} \mathcal{L}_{\text{MSE}}(v_i, v_i^{\text{gt}}) + \lambda_{\text{mask}} \mathcal{L}_{\text{MSE}}(v_i^\alpha, v_i^{\alpha\text{gt}}) + \lambda_{\text{LPIPS}} \mathcal{L}_{\text{LPIPS}}(v_i, v_i^{\text{gt}}) + \lambda_{\text{reg}} \mathcal{L}_{\text{reg}}, \quad (5)$$

where $\mathcal{L}_{\text{MSE}}$ and $\mathcal{L}_{\text{mask}}$ represent the mean square error loss in the RGB image and the alpha mask, respectively; $\mathcal{L}_{\text{LPIPS}}$ represents the well-adopted VGG-based perceptual loss [70] ; $\mathcal{L}_{\text{reg}}$ is the opacity L1 regularization loss $||1 - \alpha_i||$ encourage more efficient use of each Gaussian by enforcing higher density. $\lambda_{\text{mask}}$, $\mathcal{L}_{\text{LPIPS}}$ and $\lambda_{\text{reg}}$ are the trade-off coefficients that balance each loss.

### 3.4 Unified 3D Generation Inference

During inference (Figure 3(b)), the pre-trained reconstructor can be smoothly combined with any off-the-shelf multi-view diffusion models to efficiently predict a set of Gaussians, which facilitates both text-to-3D and image-to-3D generation. Typically, we leverage ImageDream [13] and MVDream [14] to produce 4 multi-view images with anchored poses [11] from a single image or text prompt, respectively. For mesh extraction, following Huang et al. [62], we utilize truncated signed distance fusion (TSDF) [71] that fuse the depth maps rendered from the output Gaussians to obtain a smooth polygonal mesh.

## 4 Experiment

### 4.1 Experimental Settings

**Training dataset.** We obtain the multi-view images from Objaverse [7] for MVGamba pre-training. Following [19, 72], we filtered $80k$ valid high-quality 3D objects. We then used Blender under uniform lighting to render 25 views of RGBA images with their alpha masks at a resolution of $512 \times 512$, in the elevation range of $5°$ to $30°$ with rotation $\{15° \cdot r | r \in [0, 23], r \in \mathbb{N}\}$. To align with the camera configurations in ImageDream [13] and MVDream [14], at each training step, we select 4 images of a certain object as input views with the same elevations, while rotations separated by $90°$, denoted as $\{\phi + 90° \cdot k \mid k \in 0, 1, 2, 3\}$ and another random set of 6 views as supervision.

**Implementation details.** MVGamba is trained on 32 NVIDIA A100 (80G) with batch size 512 for about 2 days. We adopt gradient checkpointing and mixed-precision training with BF16 data type to ensure efficient training and inference. We use the AdamW optimizer with learning rate $1 \times 10^{-3}$ and weight decay 0.05, following a linear learning rate warm-up for 15 epochs with cosine decay to $1 \times 10^{-5}$. The output Gaussians are rendered at $512 \times 512$ resolution for mean square error loss and resized to $256 \times 256$ for LPIPS loss for memory efficiency. The trade-off coefficients that balancing each loss were set as $\lambda_{\text{mask}} = 1.0$, $\lambda_{\text{LPIPS}} = 0.6$ and $\lambda_{\text{reg}} = 0.001$. We also follow the common practice [19] to clip the gradient with a maximum norm of 1.0. The detail of MVGamba model configuration is included in *Appendix D*.

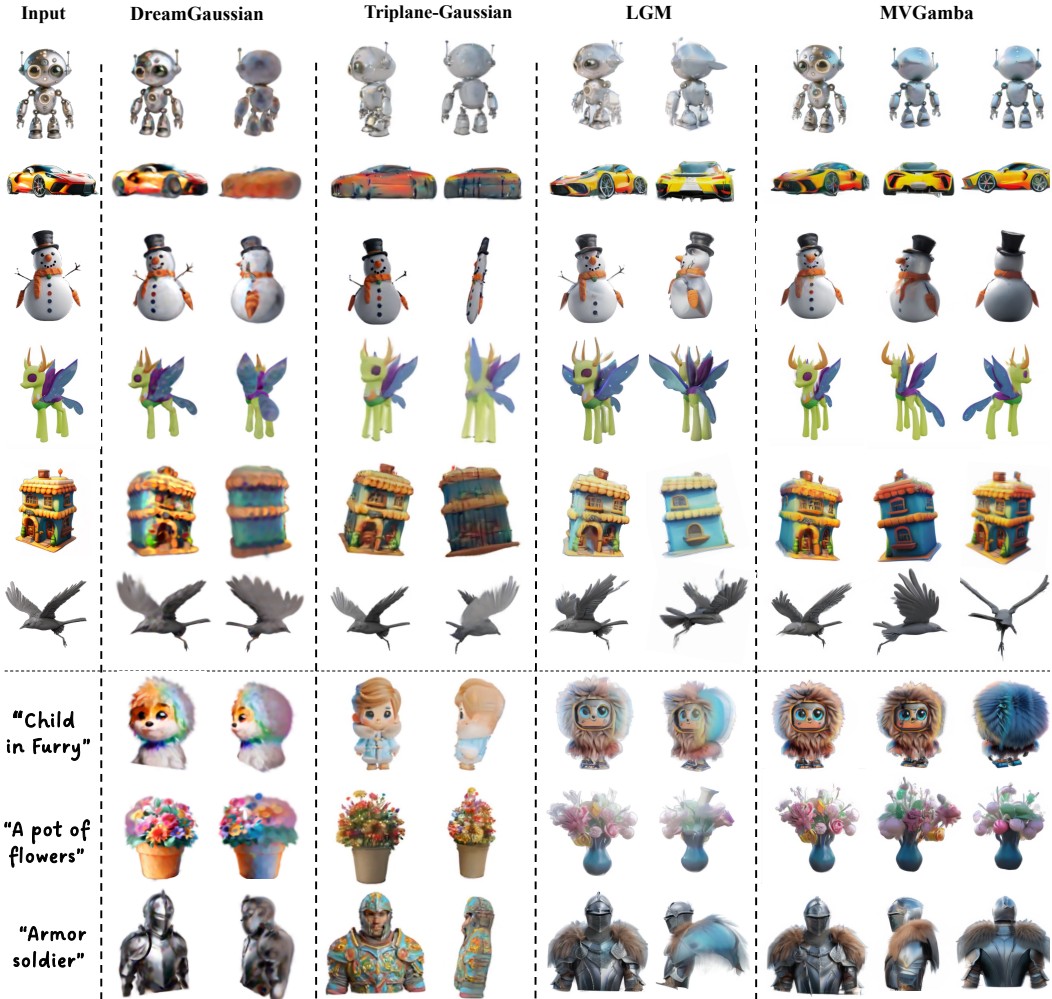

| Input | DreamGaussian | Triplane-Gaussian | LGM | MVGamba |
|---|---|---|---|---|

Figure 4: Qualitative comparison in image-to-3D and text-to-3D generation. Please refer to *Appendix* C for more generation results.

## 4.2 Comparison against Baselines

In this section, we compare MVGamba with previous state-of-the-art instant 3D generation methods in image-to-3D, text-to-3D generation, and sparse-view reconstruction tasks. For each task, we first elaborate on the evaluation metrics and baseline methods, then perform an extensive qualitative and quantitative comparison.

**Single image-to-3D generation.** We make comparisons to recent methods, including optimization-based DreamGaussian [34], Wonder3D [73]; feed-forward methods LGM [19], TripoSR [74] and Triplane-Gaussian [33] and One-2345++ [75]. We adopted the official codebase and pre-trained model weight for all the above methods and we are quite confident that the baselines presented are the finest re-implementations we have come across[†]. We evaluated the generation quality of MVGamba with a wide range of wild images in Figure 4. We use well-adopted PSNR, SSIM and LPIPS for quantitative measurement in the GSO [76] dataset following [18], with a total of 16 test views with equidistant azimuth and $-10 \sim 10$ degree elevations. As illustrated in Figure 4, MVGamba maintains high fidelity and plausible generation in most scenarios. In contrast, Triplane-Gaussian severely suffers from flat and blurred views, which is a notoriously ill-posed challenge, as stated by Instant3D [11]. Moreover, LGM frequently showcases multi-view inconsistency with a transparent surface, which may be attributed to its suboptimal parameter constraints and merge operation. In

---

[†]Note that Instant3D [11], GS-LRM [20] and GRM [18] are not included for comparison in the current version, as no code has been publicly released yet.

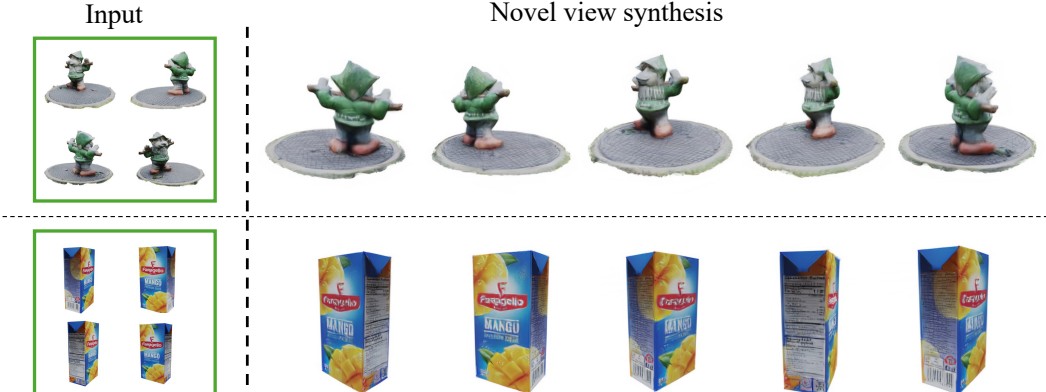

Figure 5: Qualitative results in sparse-view reconstruction. Given four views as input, MVGamba effectively reconstructs both the geometric structure and detailed textures.

Table 1: Qualitative comparison of different methods based on various metrics.

| Method | PSNR↑ | SSIM↑ | LPIPS↓ | CLIP↑ | R-Prec↑ | INF. Time↓ |
|---|---|---|---|---|---|---|
| DreamGaussian [19] | 21.59 | 0.80 | 0.16 | 0.793 | 53.5 | 120s |
| Wonder3D [73] | 22.92 | 0.84 | 0.19 | 0.850 | 49.8 | 200s |
| One-2-3-45++ [75] | 23.10 | 0.85 | 0.15 | 0.869 | 57.1 | 75s |
| TripoSR [74] | 23.76 | **0.91** | 0.09 | 0.815 | 50.1 | 0.5s |
| TriplaneGaussian [33] | 21.85 | 0.79 | 0.14 | 0.714 | 28.9 | **0.2s** |
| LGM [19] | 22.80 | 0.84 | 0.11 | 0.827 | 43.9 | 5s |
| **MVGamba (Ours)** | **24.82** | 0.90 | **0.08** | **0.895** | **60.4** | 4s |

Table 1, MVGamba outperforms other baselines in most evaluation metrics. Note that although our inference speed (4 seconds) is relatively slower than Triplane-Gaussian, we achieve significantly better generation quality by a large margin of **2.97 dB** PSNR.

**Text-to-3D generation.** Similar to the single image-to-3D task, we compare MVGamba with several optimization-based methods and feed-forward models LGM, TripoSR, and Triplane-Gaussian, using various prompts. For methods that only accept a single image as input, such as TripoSR and Triplane-Gaussian, we use DALL·E 3 [77] to transform the given text into a single image. Note that for more fair comparison, MVGamba and LGM take the same generated multi-view inputs. Figure 4 shows that MVGamba excels at generating plausible geometry and highly realistic texture. Due to its effective ray embedding and multi-view diffusion models, MVGamba is also free from the multi-face problems that frequently occur in optimization-based methods. We further randomly selected 50 prompts from the Dreamfusion [1] gallery and used CLIP Precision and CLIP scores [78–80] to measure appearance quality and alignment with the given prompt. Our high generation quality is also vividly reflected in Table 1, where MVGamba consistently ranks the highest among all baselines.

**Sparse-view reconstruction.** Given the same sparse-view inputs, we compare MVGamba with SparseNeuS [81] (trained in One-2-3-45), SparseGS [82] and LGM [19] that are capable of generating 3D Gaussians. We visualize the reconstruction results of MVGamba in Figure 5 and compare against baselines on the GSO [76] dataset with 32 randomly selected views for each object. As shown in Figure 2 and Figure 6, MVGamba is able to faithfully reproduce high-frequency details with accurate geometric modeling. Due to the limited space, the quantitative evaluation is presented in *Appendix* C.

## 5 Ablation and Discussion

**Q1:** *Why MVGamba outperforms other LRMs in most 3D content generation tasks?*

**A1:** To better diagnose the progress of MVGamba, we conduct two ablation experiments on G-buffer Objaverse [72] Human subset: one is a counterfactual experiment simulating severe inconsistency

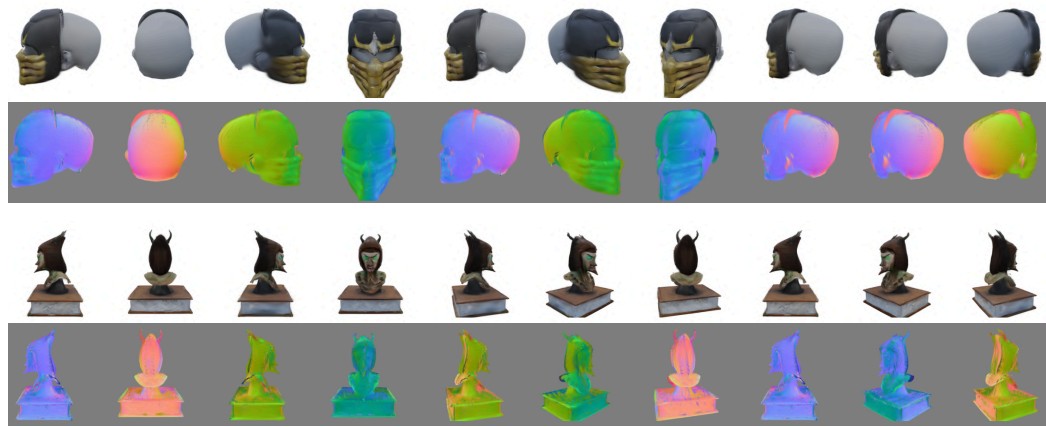

Figure 6: RGB images and normal maps rendered by MVGamba-2DGS.

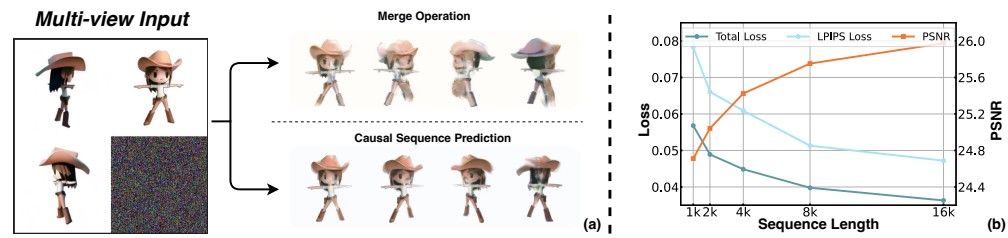

Figure 7: (a) Worst-case simulation of the inconsistency introduced by the multi-view diffusion model. (b) The effect of sequence length on 3D reconstruction.

during inference to test robustness, and the other on the effect of Gaussian sequence length for sparse-view reconstruction. **(1)** In Figure 7(a), we manually perturb one of the four input images with Gaussian noise to simulate the worst-case multi-view input inconsistency caused by the diffusion model. We then feed these manipulated images into the MVGamba reconstructor and merge-operation reconstructor for comparison. As expected, generating 3D Gaussians conditioned on each view separately and simply merging the outputs treats the perturbed input as faithfully as the other views, resulting in inconsistency and even corrupted results. In contrast, our Gaussian reconstructor generates the Gaussian sequence in a causal and self-refining manner, allowing it to mitigate the effects of the perturbation by leveraging propagated multi-view information. This experiment demonstrates that our MVGamba is highly resilient to inconsistencies in multi-view diffusion models used in the first stage due to its causal sequence modeling. **(2)** We also investigate the effect of Gaussian sequence length by varying the patch size to model different sequence lengths. As shown in Figure 7(b), our SSM-based Gaussian reconstructor can directly generate extremely long Gaussian sequences, and its performance improves with increasing sequence length. From these two aspects, we can conclude that the higher generation performance of MVGamba is indeed attributable to its self-refineable multi-view modeling and efficient utilization of sufficiently long Gaussian sequences.

**Q2:** *What impacts performance of MVGamba in terms of component-wise contributions?*

**A2:** In Table 2, we analyze the component-wise contributions of MVGamba by verifying our design choices of multi-view image encoder, Gaussian decoder structure and training strategy. Note that this ablation is conducted on the filtered subset of Human category in G-buffer Objaverse [72] using smaller model architecture for better energy efficiency. Considering symbol simplicity, we denote patchify + convluation as PC; Gaussian Decoder with RotNet as GD; stable training strategy as ST. Table 2 illustrates that the replacement or exclusion of any component from MVGamba resulted in a significant degradation

Table 2: Performance comparison of different model configurations.

| Model | PSNR↑ | SSIM↑ | LPIPS↓ |
|---|---|---|---|
| w/o PC | 25.20 | 0.827 | 0.102 |
| w/o GD | 25.41 | 0.788 | 0.096 |
| w/o ST | 26.69 | 0.910 | 0.065 |
| Ours | 27.13 | 0.925 | 0.057 |

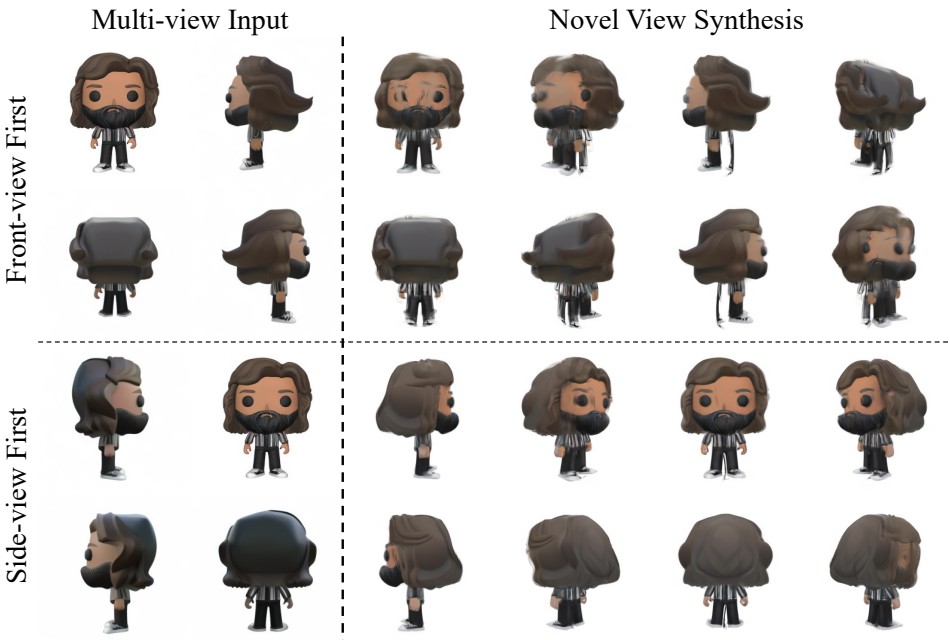

Figure 8: Ablation study on input order. Top: MVGamba may fail if the depth of the front-view is estimated incorrectly and the front-view is given first. Bottom: Manually changing the input order to provide the side-view first allows MVGamba to generate satisfactory 3D content, as the side-view contains sufficient depth information.

in performance. In particular, if we replace our designed Gaussian Decoder with the one in previous LRMs [11, 29] (10-layer, 64-width shared MLP), we notice a large performance drop due to degraded sequential modeling ability and a tendency for the deep MLP to overfit. Additionally, the lack of restrictions on position and rotation could cause more artifacts during both training and inference.

**Q3:** *What is the limitation of MVGamba?*

**A3:** Honestly, though MVGamba achieves promising results, it still has several limitations. **(1)** As we model the Gaussian sequence causally, MVGamba can sometimes fail if the depth of the front view is not estimated correctly (Figure 8 top). Fortunately, we empirically find that this limitation can be mitigated by manually changing the input order. For example, using a side-view as the first input allows our model to generate satisfactory 3D content, as the side-view contains sufficient depth information (Figure 8 bottom). In the future, we may explore automatic ways to optimize the input order to enhance robustness. **(2)** The generation quality of MVGamba is highly dependent on the four input views provided by off-the-shelf multi-view diffusion models [13, 14]. However, current multi-view diffusion models are far from perfect, known to exhibit 3D inconsistencies [9, 19, 83], and limited to a resolution of $256 \times 256$. We expect that our model's performance can be seamlessly boosted with the advancements in multi-view diffusion models in future work.

## 6 Conclusion

In this paper, we introduce MVGamba, a general and lightweight Gaussian reconstruction model for unified 3D content generation. MVGamba features a novel multi-view Gaussian reconstructor based on state space sequence modeling, maintaining multi-view information integrity and enabling cross-view self-refinement. It generates long Gaussian sequences with linear complexity in a single forward process, eliminating the need for post hoc operations. Extensive experiments demonstrate that MVGamba (49M parameters) outperforms state-of-the-art LRMs in various 3D generation tasks with only $0.1\times$ the model size. In general, MVGamba achieves state-of-the-art quality and high efficiency in parameter utilization, training, inference, and rendering speed. In the future, we aim to apply MVGamba to a wider range of 3D generation tasks, such as scene and 4D (dynamic) generation.

# 7 Acknowledgments

This project is supported by Kunlun 2050 Research, Skywork AI and Agency for Science, Technology AND Research, and by the National Research Foundation, Singapore, under its Medium Sized Center for Advanced Robotics Technology Innovation. Pan Zhou was supported by the Singapore Ministry of Education (MOE) Academic Research Fund (AcRF) Tier 1 grants (project ID: 23-SIS-SMU-028 and 23-SIS-SMU-070).

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

# Appendices

The *Appendix* is organized as follows:

- **Appendix A:** elaborates more details about our MVGamba pipeline. Specifically, we detailed the implementation of inference pipeline and mesh extraction.
- **Appendix B:** discusses the comparison between concurrent transformer-based methods and the function or pixel-aligned representation.
- **Appendix C:** showcases more qualitative and quantitative experiment results.
- **Appendix D:** provides further details on model design and configuration.

## A    Implementation Details

**Inference.**    For single image input, MVGamba first segments the recentered foreground object using a pre-trained segmentation model [84], then leveraging a multi-view diffusion model, typically ImageDream, to generate four posed views as input for the SSM-reconstructor. Similarly, for text prompt input, MVGamba uses MVDream to transform the given text into four posed views. For sparse-view reconstruction, MVGamba takes the given few views as the ground-truth multi-view inputs. MVGamba employs a default camera pose with both zero elevation and azimuth to produce camera tokens and Plücker ray inputs. Remarkably, this process requires only about *10 GB* of GPU memory (for both the multi-view diffusion model and the Gaussian Reconstructor) and completes in less than *5 seconds* (4.5 seconds for multi-view image generation and 0.03 second for predicting Gaussians and real-time rendering) on a single NVIDIA A800 (80G) GPU, making it well-suited for online deployment scenarios.

**Mesh extraction.**    For mesh extraction from reconstructed 2D splats, following [32], we render depth maps of the Gaussian rendering views by using the median depth value of the splats projected onto the pixels. We then use truncated signed distance fusion (TSDF) with Open3D to fuse the reconstruction depth maps. We reset the voxel size and the truncation threshold during TSDF fusion suitable for object mesh extraction.

## B    More Discussion

### B.1    Comparison with concurrent works.

We illustrate the two current sequence modeling paradigms for Gaussian LRMs: Compressed vs. Direct Sequence Modeling (ours) in Figure 9. Below, we elaborate on these two paradigms to highlight the unique approach of our proposed paradigm compared to recent transformer-based approaches, *e.g.*, GRM, GS-LRM.

- (a) **Compressed Sequence Modeling:** This approach tokenizes the input into a compact representation, processes it through a series of transformer blocks, and then upsamples to produce the 3DGS parameters. This paradigm is represented by the concurrent GS-LRM and GRM.
- (b) **Direct Sequence Modeling (ours):** This approach tokenizes and expands the input into a sufficiently long sequence of tokens through cross-scan operations, which are then directly processed by a series of mamba blocks to generate the 3DGS parameters.

Our proposed paradigm (b) offers several significant benefits:

- **Efficient long sequence modeling:** It accommodates larger spatial dimensions to capture and preserve fine-grained details while mitigating information loss typically caused by upsamplers, such as the zero-padding in de-convolution layers. This benefits accurate geometry and texture reconstruction. As demonstrated in Table 3, Mamba's linear computational complexity allows for a favorable balance between computational cost and long-sequence modeling capacity, enabling efficient processing of high-resolution inputs without prohibitive memory requirements.

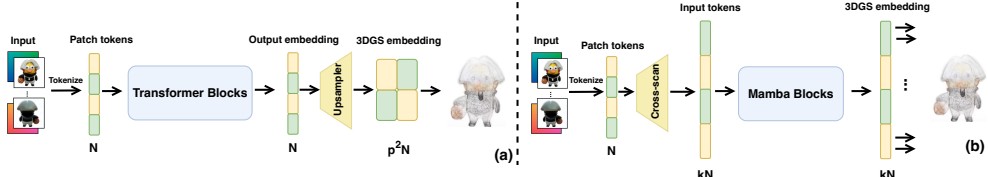

Figure 9: The illustration of two type of Gaussian LRMs. (a) Compressed sequence modeling (b) Direct long-sequence modeling (ours).

Table 3: Theoretically calculated FLOPs comparison between self-attention in Transformer and SSM in Mamba. Here, dimension $D = 512$; GFLOPS of both modules are calculated according to Equations (5) and (6) in Vision Mamba [26].

| Model / Length | 1024 | 2048 | 4096 | 8192 | 16384 | 32768 |
|---|---|---|---|---|---|---|
| Self-attention / GFLOPs | 2.15 | 6.44 | 21.47 | 77.31 | 292.06 | 1133.87 |
| SSM / GFLOPs | 0.07 | 0.13 | 0.27 | 0.54 | 1.07 | 2.15 |

- **Easy of optimization:** It directly models the 3DGS sequence without any upsampler, hence establishing a more straightforward relationship between 3DGS parameters (e.g., position, color) and the modeled sequence. This eases the non-convex optimization in inverse graphics scenario, as discussed in PixelSplat, and helps learn accurate geometry and textures.

- **Cross-view self-refinement:** It efficiently incorporates multi-view information causally from the initial condition onward, thereby enabling the refinement of inconsistent parts based on earlier views and generated tokens.

## B.2 The function of pixel-aligned Gaussian

We conducted an additional experiment using a non-pixel-aligned transformer model for 3DGS prediction. As discussed in Appendix B.1, transformer-based methods typically process a compressed sequence considering the computational budget. We then followed OpenLRM to model 4096 tokens, and adopt a de-convolution layer to up-sample the 3DGS tokens from 4096 to 16384, similar to TGS for intricate 3D modeling. However, as shown in Figure 11, the non-pixel-aligned transformer yielded inferior performance, with broken geometry and extremely blurred textures after 300 epochs of training. We attribute this to the inherent non-convex optimization challenge in the inverse graphics scenario, as mentioned in pixelSplat [85], which may be further amplified by the upsampler. This preliminary result may also explain why pixel-aligned approaches are proposed and are adopted by recent transformer-based approaches with upsamplers. On the other hand, our proposed direct sequence modeling paradigm allows for more fine-grained detail modeling and exhibits a cross-view self-refinement ability. Moreover, our direct sequence modeling paradigm can also ease optimization, which provides a new feasible way for training Gaussian LRMs.

## C  More Results

Table 4: Quantitative comparison on sparse-view reconstruction.

| Method | #views | PSNR↑ | LPIPS↓ | SSIM↑ | INF. Time↓ | CD↓ | VIoU↑ |
|---|---|---|---|---|---|---|---|
| SparseGS [82] | 16 | 22.19 | 0.162 | 0.775 | 34s | - | - |
| SparseNeuS [81] | 16 | 23.17 | 0.130 | 0.814 | 6s | 0.0566 | 0.3479 |
| LGM [19] | 4 | 24.20 | 0.112 | 0.845 | 0.07s | 0.0198 | 0.4410 |
| MVGamba | 4 | **26.25** | **0.069** | **0.881** | **0.03s** | **0.0132** | **0.4829** |

**More qualitative results.**    We include more qualitative experiment on image-to-3D generation with rendering results in Figure 10 for novel view synthesis and in Figure 12 for normal map generation.

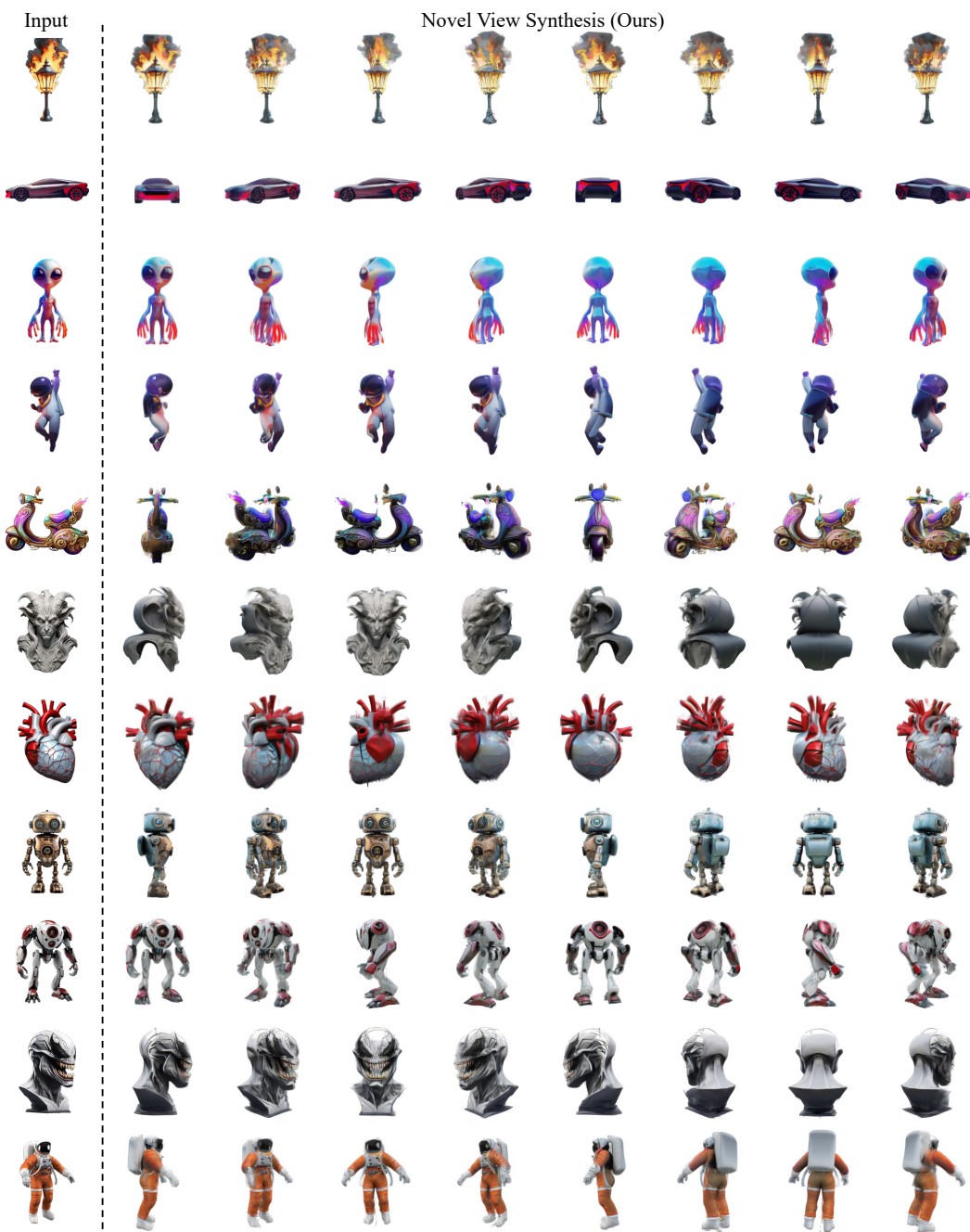

Figure 10: More qualitative results of MVGamba on image-to-3D generation.

We also include a qualitative comparison using samples from the GRM paper in Figure 13. It indicates that MVGamba captures better geometries and details with significantly smaller model size. Note that we could not directly compare with GS-LRM as it only presents a single-view results in Figure 7 of their paper.

**More quantitative results.**    We leave the qualitative comparison on sparse-view reconstruction task here. We compare the well-adpoted PSNR, SSIM, LPIPS for novel view synthesis [86] and Chamfer Distance (CD) and Volume IoU (VIoU) for geometric evaluation. As shown in Table 4, MVGamba outperforms all baselines across all metrics, even though SparseNeuS require 4 times more input views, while maintaining fast inference and rendering speed.

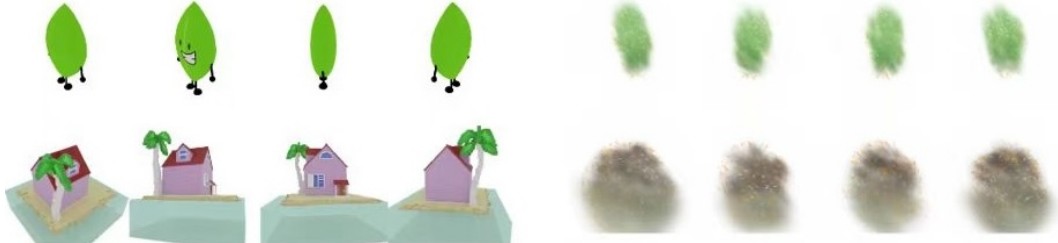

Figure 11: The generation results of non-pixel-aligned transformer-based Gaussian LRM, resulting in extremely broken geometry and blurred texture.

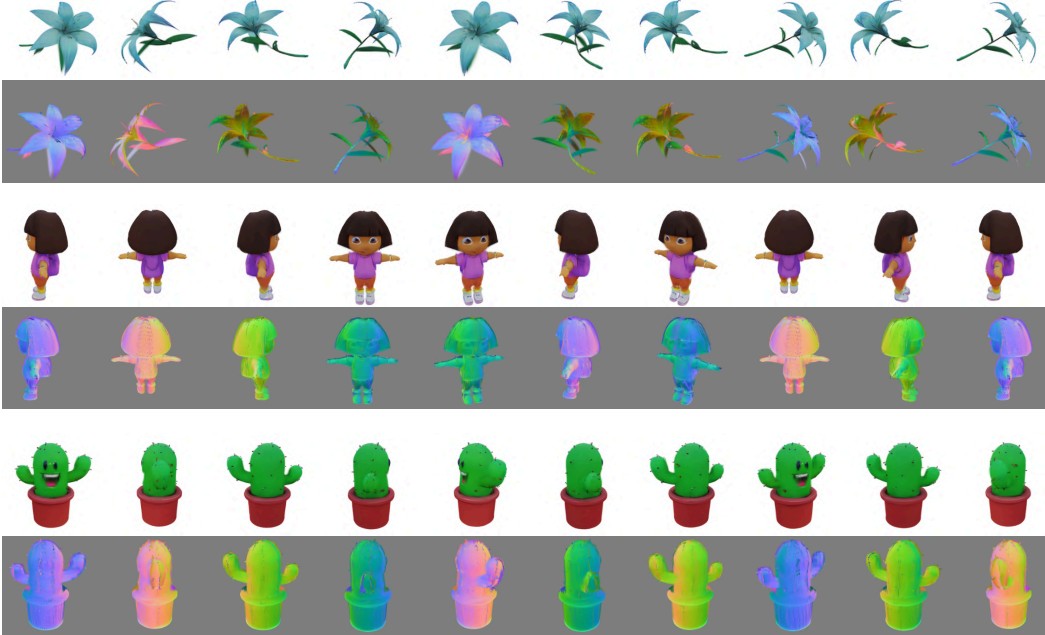

Figure 12: RGB images and normal maps rendered by MVGamba-2DGS.

## D  Model Details

**Network Configuration.**  As detailed in Table 5, the SSM-based reconstructor comprises 14 Gamba blocks, each with hidden dimensions of 512. The architecture employs RMSNorm, SSM, depth-wise convolution [87], and residual connections. In line with [24], positional embedding is added rather than concatenated to the tokenized multi-view image tokens. These tokens have 512 dimensions, resulting in a total of 16,384 or 32,768 tokens, which correspond to the length of Gaussian sequence. The Gaussian Decoder is a compact multi-layer perceptron (MLP) with a single hidden layer containing 2,048 hidden dimensions. This is followed by sub-head linear projection layers, which decode the output into 3D Gaussians for splatting.

**RotationNet.**  Our RotationNet (RotNet) is designed to facilitate the prediction of rotation attributes. It comprises a set of 32 pre-defined canonical rotation quaternions and a learnable linear projection matrix that maps input features to the logits of these quaternions. The canonical quaternions include 8 rotations around the principal axes $(x, y, z)$ by 0 or 180 degrees, and 24 rotations by $\pm 45$ degrees around various axes formed by combining two principal axes. During training, RotNet uses the Gumbel-Softmax technique to enable differentiable sampling from the discrete set of quaternions. The temperature parameter is gradually decreased from 2 to 0.01 over iterations, encouraging more confident predictions. During inference, RotNet applies the argmax operation to the logits to select the quaternion with the highest probability, operating in a non-differentiable manner.

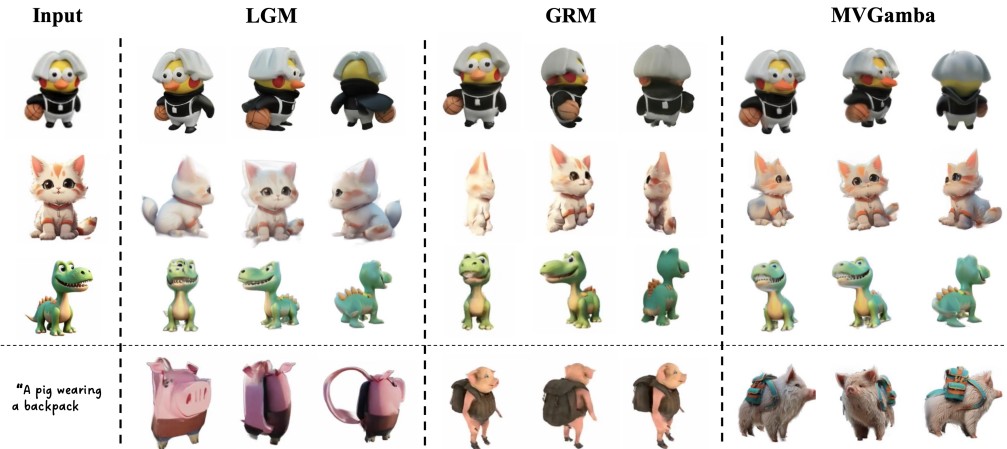

Figure 13: More comparisons with concurrent state-of-the-art large Gaussian models. Note that the generation results for GRM were extracted from its paper and official project page. A direct comparison with GS-LRM is currently unfeasible, as it is not open-source, and only a single view is presented in Figure 7 of the GS-LRM paper.

---

**Algorithm 1:** Gaussian parameter constraint

```
# Input: feats: [b, n, ...]
# Output: gsparams: list of constrained parameters

constraints = {
    "xyz": lambda v: (F.normalize(v.reshape(*v.shape[:2], 3,
        -1), dim=-1) * coords[None, None]).sum(dim=-1),

    "scale": lambda v: 0.1 * F.softplus(v),

    "rot": lambda v: RotationNet()(v),   #proposed RotNet

    "opacity": lambda v: torch.sigmoid(v),

    "rgb": lambda v: torch.sigmoid(v),

    "shs": lambda v: v   # No apply SHS
};
```

Figure 14: The pseudo code for Gaussian parameter constraint.

**Gaussian Parameterization.** As 3D Gaussians are an unstructured explicit representation, unlike Triplane-NeRF's structured implicit representation, the parameterization of the output parameters significantly affects the model's convergence. We provide the detailed pseudo code of Gaussian parameterization in Figure 14 for better reprociblility.

# E   Licenses

Datasets:

- Objaverse [7]: ODC-By v1.0 license

Pre-trained models:

Table 5: Detailed model configuration of MVGamba.

| Parameter | Value |
|---|---|
| **Image Tokenizer** | |
| **image resolution** | $448 \times 448$ |
| **patch size** | 14 |
| **# channels** | 512 |
| **Backbone** | |
| **Mamba layers** | 14 |
| **# channels** | 512 |
| **state expansion factor** | 16 |
| **local convolution width** | 4 |
| **block expansion factor** | 2 |
| **normalization** | RMSNorm |
| **Decoder MLP** | |
| **width** | 2048 |
| **# hidden layers** | 1 |
| **activation** | SiLU |
| **Training** | |
| **optimizer** | AdamW |
| **epochs** | 300 |
| **batch size** | 512 |
| **learning rate** | 1e-3 |
| **weight decay** | 0.05 |
| **gradient clipping** | 1.0 |
| **Adam** $(\beta_1, \beta_2)$ | $(0.9, 0.95)$ |
| **lr scheduler** | CosineAnnealingLR |
| **# warm-up epochs** | 15 |
| $\lambda_{\text{mask}}$ | 1.0 |
| $\lambda_{\text{LPIPS}}$ | 0.6 |
| $\lambda_{\text{reg}}$ | 0.1 |

- ImageDream [13]: Apache-2.0 license
- MVDream [14]: MIT License
- SAM [84]: Apache-2.0 license

