# OpenReview forum: "MVGamba: Unify 3D Content Generation as State Space Sequence Modeling"
_NeurIPS.cc/2024/Conference — NeurIPS 2024 poster_

### Official Review · Reviewer_5iTN · 2024-06-16

**Soundness:** 3
**Presentation:** 3
**Contribution:** 3
**Rating:** 6
**Confidence:** 4

**Summary:**

This work introduce MVGamba, a general and lightweight Gaussian reconstruction model featuring a multi-view Gaussian reconstructor based on the RNN-like State Space Model (SSM). The Gaussian reconstructor propagates causal context containing multi-view information for cross-view self-refinement while generating a long sequence of Gaussians for fine-detail modeling with linear complexity. With off-the-shelf multi-view diffusion models integrated, MVGamba unifies 3D generation tasks from a single image, sparse images, or text prompts. Extensive experiments demonstrate that MVGamba outperforms state-of-the-art baselines in all 3D content generation scenarios with approximately only 0.1x of the model size.

**Strengths:**

1. The paper is of good written quality.
2. The inference speed is relatively fast compared to optimization-based methods.
3. This work introduces the new architecture, Mamba, into the field of 3D generative models, which is novel.
4. The computation cost of the training is lower than LGM and LRM.
5. The method is robust to multiview inconsistency.

**Weaknesses:**

1. In Fig. 6(b), the performance is continuously increasing as the token length increase. Although the authors' motivation is to lower the computation cost, Fig. 6(b) indicates the the performance grows as the computation cost grows.
2. No failure cases are shown to demonstrate the limitation of this work.
3. Fig 2(b) may be wrong. Mamba has linear complexity instead of constant complexity.
4. The video results in the supplementary materials are only marginally better than LGM's results.

**Questions:**

See the weaknesses.

**Limitations:**

See the weaknesses.

---

> ### Author Rebuttal · Authors · 2024-08-07
>
> Thank you for your insightful and positive comments! Below, we provide a point-by-point response to address your concerns. We welcome further discussion to improve the clarity and effectiveness of our work.
>
>
> > **Q1: The performance is continuously increasing as the token length increase.**
>
>  **A1:** The experimental results in Figure 6(b) confirm that performance improves as the token length increases. This is one of our motivation to *directly* process a sufficiently long token length for better performance. Consequently, we advocate for a Mamba-based architecture, which boasts linear complexity, over transformer-based architectures characterized by quadratic complexity to properly **balance** the performance and the computational cost, as illustrated in Table 1 in the rebuttal PDF. Furthermore, our architectural design incorporates various light-weight design elements, as described in the manuscript and detailed in *Appendix* C, to further reduce the computational cost and energy consumption, thereby promoting environmental sustainability in line with the growing emphasis on green AI in the Neurips society.
>
>  >**Q2: No failure cases are shown to demonstrate the limitation of this work.**
>
>  **A2:**
>
> - **We have illustrated the failure cases in the *Appendix* D.** As shown in Figure 8, MVGamba may sometimes fail if the depth of the front view is estimated incorrectly. Fortunately, this issue can be largely mitigated by manually changing the input order, as the side view typically contains sufficient depth information.
>
> - **Due to limited space in the rebuttal PDF, we couldn't provide additional failure cases discussed in the *Limitations* section**, particularly those caused by severely flawed input views generated by the imperfect multi-view diffusion model. It's worth noting that, compared to other baselines, our results demonstrate greater robustness against inconsistent inputs due to the self-refining nature of MVGamba, which aligns with the experimental results presented in Figure 6(a) of the manuscript. We will incorporate more failure cases and their analysis in the revision as per your advice.
>
> >**Q3: Fig 2(b) may be wrong. Mamba has linear complexity instead of constant complexity.**
>
>  **A3:** We apologize for the confusion. We confirm that Figure 2(b) is correct; it indeed illustrates linear complexity. Due to the quadratic complexity of the Transformer, we had to scale the y-axis to prevent the Transformer's curve from extending outside the figure. We will revise this figure to avoid confusion in revision.
>
> > **Q4: The video results in the supplementary materials are only marginally better than LGM's results.**
>
>  **A4:** We believe that our proposed MVGamba could consistently outperform LGM in most cases. To address your concerns, we have provided more qualitative results compared to LGM in Figure 3 in the rebuttal PDF, where MVGamba could generate more accurate geometry and fine-grained textures, while LGM exhibits blurred texture and severe multi-view inconsistency, eg., multiple eyes on the dragon's face.

---

> > ### Comment · Reviewer_5iTN · 2024-08-11
> >
> > Thank you for your rebuttal. My concerns have been partially solved and I keep my score. I hope the Fig 2(b) can be corrected in further revision of this work.

---

> ### Author Response · Authors · 2024-08-11
> **Thanks for the response and support!**
>
> Dear Reviewer 5iTN,
>
> Thank you so much for your response and support. Regarding Figure 2, we observe that as the token length increases from 1024 to 16384, the theoretical FLOPs for transformer self-attention rise dramatically from 2.15 to 292.06, while Mamba's SSM demonstrates a linear increase in FLOPs, scaling from 0.07 to 1.07 as shown in Table 1 of the rebuttal PDF. Plotting these together necessitates scaling the y-axis, which could potentially cause confusion by making Mamba's complexity **appear constant, which indeed is linear**.  Per your suggestion, we will clearly **indicate the y-axis scale** and **specific FLOPs values** to prevent misinterpretation in the revision.
>
> If you have any further concerns or questions, welcome to discuss with us. We are more than happy to discuss with you further and provide additional materials.
>
> Once again, thank you for your valuable time and sound advice!
>
> Best regards,
>
> Authors of Submission 2178

---

### Official Review · Reviewer_eshR · 2024-07-12

**Soundness:** 3
**Presentation:** 3
**Contribution:** 2
**Rating:** 6
**Confidence:** 4

**Summary:**

This paper introduces MVGamba, a feed-forward, sparse reconstruction model. This model takes a small number of images (e.g., 4 views) to infer 3D Gaussians. Basically, it is a Mamba version of multi-view large reconstruction model. The authors have implemented several strategies to ensure stable training and optimal performance.

Experimental results demonstrate that the MVGamba model can reconstruct 3D Gaussians with higher quality than the Large Gaussian Model (LGM).

**Strengths:**

(1)	The experimental results are somewhat promising, showing better Gaussian reconstruction quality than LGM, although the quality is not as high as the concurrent GRM and GS-LRM works.

(2)	I like the experiment design in Section 5 Q1; it provides some motivation for why we need a non-pixel-aligned architecture. It seems this kind of architecture is more robust to multi-view inconsistency.

(3)	The paper is clearly written and easy to understand.

**Weaknesses:**

(1)	In the abstract and also in Lines 40-41, the authors state that the generated 3D models often suffer from multi-view inconsistency and blurred textures, which is why MVGamba is proposed. However, it seems that this issue is more likely due to whether the architecture is pixel-aligned or not. Therefore, I believe a non-pixel-aligned transformer model can also address these issues. Did the authors try this? How does a non-pixel-aligned transformer model perform?

(2)	I cannot find any tables or numbers in the paper to support the conclusion “0.1× of the model size.” Did the authors forget to include them?

(3)	The reconstruction results look in lower quality than the concurrent GRM and GS-LRM works.

**Questions:**

(1)	Line 53: Could it be more specific? What does 'post hoc' operation specifically mean?

(2)	What is the number of output 3D Gaussian primitives? Does it equal N (the number of tokens)? Is there any way to increase the number of Gaussians for better reconstruction quality?

(3)	Given the advantages of Mamba in computational efficiency for long sequences, did the authors try to train an MVGamba with a larger number of views, such as 10 views?

**Limitations:**

The method is limited in object-level reconstruction with clean background. I encourage the authors explore on scene-level reconstruction as an interesting future work.

---

> ### Author Rebuttal · Authors · 2024-08-07
>
> Thank you for your positive feedback and insightful comments! Below, we provide a response to address your concern. We welcome further discussion to enhance the clarity and effectiveness of our work.
>
> >**Q1: How does a non-pixel-aligned transformer model perform?**
>
>  **A1:**
>
> - **Non-pixel-align is not the primary cause.** Per your suggestion, we tested a non-pixel-aligned transformer model for 3DGS prediction. As discussed in our overall response, those methods typically process a compressed sequence due to the computational budget. We followed OpenLRM[1] to model 4096 tokens, and adopt a de-convolution layer to up-sample from 4096 to 16384. However, as shown in the rebuttal PDF Figure 2, it yielded inferior performance, with broken geometry and blurred textures. We attribute this to the inherent non-convex optimization challenge in the inverse graphics scenario, as mentioned in pixelSplat[2], which may be further amplified by the de-conv upsampler. This preliminary result may also explain why pixel-aligned Gaussian are widely adopted by recent transformer-based approaches with upsamplers.
>
> - **The crux is to directly model a sufficiently long sequence of 3DGS in a causal manner.** As mentioned in the manuscript and our overall response, the direct sequence modeling paradigm allows for more fine-grained detail modeling with cross-view self-refinement . Moreover, our direct sequence modeling paradigm can also ease optimization, which provides a new feasible way for training Gaussian LRMs.
>
> [1] He, Z. and Wang, T. OpenLRM: Open-Source Large Reconstruction Models.
>
> [2] Charatan, D., et al. pixelsplat: 3d gaussian splats from image pairs for scalable generalizable 3d reconstruction. CVPR2024.
>
> > **Q2: Model size comparison.**
>
>  **A2:** Thank you for pointing out! The following table discusses the model size that supports our claim. Our statement of *0.1× of the model size* refers to a comparison with open-source SOTA models TripoSR and LGM. This table will be included in the revision.
>
>
>
> | Model      | OpenLRM | TGS   | TripoSR | LGM   | GS-LRM | **MVGamba** |
> |:-----------:|:-------:|:-----:|:-------:|:-----:|:------:|:-----------:|
> | Model Size  |  366M   | 260M  |  ~400M  | 415M  |  300M  |   **49M**   |
>
>
>  >**Q3: Reconstruction results comparison.**
>
>  **A3:** To address your concerns, we provide an additional qualitative comparison using samples derived from the original GRM paper. As shown in the rebuttal PDF Figure 3, MVGamba accurately captures better geometric structures and comparable appearance details compared to GRM. We could not directly compare with GS-LRM as it only presents a single-view results in Figure 7 of their paper.
> Note that while GRM and GS-LRM bring impressive generation results, they were both pre-printed on arXiv close to the NeurIPS deadline without open-source code, and even **some of the multi-view diffusion models they leveraged is not available to public yet, eg., Instant3D**. Additionally, both GS-LRM, with 300M parameters, and GRM, whose parameters are not disclosed, appear significantly larger than MVGamba with 49M parameters.
>
>  >**Q4: What does 'post hoc' operation mean?**
>
>  **A4:** **'Post hoc' refers to the merge operation discussed in lines 39-40.** For instance, GS-LRM and LGM increase the number of Gaussians by simply merging from different views. In contrast, our MVGamba can directly generate a sufficiently long Gaussian sequence for all multi-views simultaneously without such operation.
> Furthermore, as discussed in our overall response, recent transformer-based methods such as GRM and GS-LRM involve an upsampler to increase the number of Gaussians *after modeling the patch tokens*. This type of post sequence modeling operation is also not required for MVGamba.
>
>  >**Q5: What is the number of output 3D Gaussians? How to increase it?**
>
> **A5:**
>
> - As shown in Figure 1(b) in the rebuttal PDF, the number of predicted Gaussians in MVGamba *equals* the number of output tokens since MVGamba directly decodes  Gaussians from the generated long Gaussian tokens.
>
> - There are several feasible ways to increase number of Gaussians.  (1) Different scan orders: Due to the causal nature of MVGamba, we can apply various scan orders to expand patches, each increasing the number of tokens as well as the number of Gaussians by N. (2) Pixel-aligned Gaussian representation. Following the concurrent Gaussian LRMs, we could leverage upsampler to unpatchify the output tokens into per-pixel Gaussians. We believe this method might be more effective with higher image resolution and more accurate and consistent multi-view inputs.
>
> >**Q6: Larger number of input views.**
>
> **A6:** We have conducted experiments with dense view settings in the sparse view task. The Table below indicates that MVGamba effectively manages 6 input views due to its computational efficiency, achieving a notable improvement in reconstruction performance in the GSO test set. Further studies with denser input views (10 views or more) will be included in the revision.
>
> | Method  | #views |  PSNR↑  | LPIPS↓ |  SSIM↑  |
> |:-------:|:------:|:-------:|:------:|:-------:|
> | MVGamba |   #4    | 26.25   | 0.069  | 0.881   |
> | MVGamba |   #6    | 27.55   | 0.060  | 0.902   |
>
>
>
> > **Q7: Scene-level reconstruction is an interesting future work.**
>
>  **A7:** Due to time limitations, we were unable to demonstrate MVGamba's capacity for scene-level reconstruction during the rebuttal period. However, we note that MVGamba's computational efficiency and cross-view self-refinement gives it potentially significant advantages  in scene reconstruction, which require more Gaussians and higher image resolution. Theoretically, with the introduced pixel-aligned Gaussian representation, MVGamba can operate at a resolution of 2560 $\times$ 1600, compared to  512 $\times$ 904 for GS-LRM, within a similar overall computational budget. We plan to further explore the Mamba-based reconstructor for scene reconstruction as future work.

---

> ### Author Response · Authors · 2024-08-13
>
> Dear Reviewer eshR,
>
> We wish to convey our sincere appreciation for your insightful and invaluable feedback, which has been of great help to us. As the discussion deadline approaches, we are keenly anticipating any additional comments or suggestions you may have. Ensuring that the rebuttal aligns with your suggestions is of utmost importance. We are deeply grateful for your commitment to the review process and your generous support throughout.
>
> Best regards,
>
> Authors of Submission 2178

---

> > ### Comment · Reviewer_eshR · 2024-08-13
> > **Comment**
> >
> > Thank you authors! The rebuttal addressed my concerns.

---

> > > ### Author Response · Authors · 2024-08-13
> > > **Thanks for the response and support!**
> > >
> > > Dear Reviewer eshR,
> > >
> > > We sincerely appreciate your  response and support. We are delighted to learn that our rebuttal has successfully addressed your concerns.
> > >
> > > Thank you for your time and valuable feedback.
> > >
> > > Best regards,
> > >
> > > Authors of Submission 2178

---

### Official Review · Reviewer_XU7f · 2024-07-13

**Soundness:** 3
**Presentation:** 3
**Contribution:** 3
**Rating:** 5
**Confidence:** 5

**Summary:**

The authors propose MVGamba, a general and lightweight Gaussian reconstruction model for unified 3D content generation. By replacing the previous LRM work’s transformer architecture with the recent model Mamba. MVGamba can generate long Gaussian sequences with linear complexity in a single forward process, eliminating the need for post hoc operations. Experiments demonstrate that MVGamba outperforms the current open-sourced LRM work in various 3D generation tasks with a smaller model size( 0.1× ). The authors also claim the proposed model has the potential to be applied to scene-level and 4D generation.

**Strengths:**

1. The paper is well-written and easy to understand, and the experiments are sufficient and well-analyzed.
2. The motivation of this paper is sound. The current LRMs papers have achieved impressive results in 3D generation, but their architecture is highly computationally intensive, especially when handling long-sequence inputs. The recent Mama has the potential to solve it.
3. The proposed pipeline achieves better results than the selected baselines.

**Weaknesses:**

1. The paper’s experiments do not align with its claims regarding the inefficiencies of transformer-based LRMs in handling long-sequence inputs. The primary issue highlighted is that these transformer-based models are not efficient for long-sequence input. However, most experiments in the paper are for single or sparse image input 3D generation. In this context, the Mamba-based architecture does not demonstrate significant speed improvements over transformer-based architectures because the input tokens are relatively short. Table 1 shows no obvious speed enhancement compared to other feedforward methods, supporting this point. Therefore, I question the necessity of Mamba for most tasks discussed in the paper. The Mamba-based reconstruction model appears to be more suitable for dense view reconstruction tasks.

2. The performance of paper doesn’t achieve the state of the art. Although in Table 1, the paper achieves better quantitative results over all baselines, some previous LRMs papers have better performance according to their paper. For example, GS-LRM attains 8 dB higher PSNR than LGM according to its table. 1, while MVGamba is just 2 dB higher than LGM. I can understand this paper didn’t compare with GS LRM because GS LRM doesn’t release its code. I am just not sure if Mamba is important for quality improvement as claimed by the authors, when some transformer-based LRMs can also achieve good (or even better) quality.

**Questions:**

- In lines 38-41, the authors state that previous Gaussian-based LRMs fail to achieve multi-view consistent results because they rely on local or mixed attention across limited multi-view image tokens or process each view separately before merging the predicted Gaussians. However, some prior works, like GS LRM, employ self-attention on all input image tokens. Despite merging the Gaussians afterward, they are predicted collectively. It is unclear why this approach leads to multi-view inconsistency.
- In lines 281-286, the authors note that performance improves with increasing sequence length, and they model different sequence lengths by varying the patch size. If decreasing patch size enhances final performance without significantly impacting efficiency, as the authors claim, why didn't they choose a smaller patch size than 14, as indicated in the appendix? 14 is actually bigger than some transformer-based LRM’s patch size. (For example, GS LRM chose 8.)

**Limitations:**

The limitations have been well discussed.

---

> ### Author Rebuttal · Authors · 2024-08-06
>
> Thank you for the positive and insightful comments! In the following, we provide our point-by-point response and look forward to the subsequent discussion.
>
> >**Q1-1: Input tokens are short.**
>
> **A1-1:  3DGS reconstruction is a long-sequence task.** We analyze the token length of two paradigms in rebuttal PDF Figure 1:
> + Direct long-sequence modeling: Requires directly processing more than 16k tokens.
> + Compressed sequence modeling: For a standard $384 \times 384$ resolution input with a large hidden dimension $D=1024$ in GS-LRM, [1] proposed a ratio-based metric, where we derive $r_L \approx 1.5 > 1$, confirming it as a long-sequence task even with compressed sequences. Note that $r_L \approx 0.04$ in traditional image classification tasks.
>
> Moreover, we agree that Mamba-based reconstructor may potentially offer more advantages as views and sequence length increase, which we consider an interesting direction for future work.
>
> [1] Yu, W., \& Wang, X. MambaOut: Do We Really Need Mamba for Vision?. arXiv:2405.07992.
>
> >**Q1-2: Inference speed in Table 1.**
>
> **A1-2:**
> + In Table 1, MVGamba's inference time includes both multi-view image generation and **multi-view reconstruction (0.03s)**, as detailed in Appendix A. When considering the reconstruction module alone, **MVGamba is at least $3.6\times$ faster** than GRM (0.11 sec), TGS (0.2 sec), and GS-LRM (0.23 sec).
>
> + Importantly, MVGamba's direct sequence modeling achieves significant inference speed improvement even when modeling sequences that are $4\times$ longer.
>
> >**Q1-3: Necessity of Mamba.**
>
> **A1-3: Mamba-based reconstructor is necessary.** Given the above responses in A1-1 and A1-2, we believe: 1) Sparse-view 3DGS reconstruction is a long-sequence task; 2) MVGamba achieves at least $3.6\times$ faster inference speed while directly modeling $4\times$ longer sequences. To our knowledge, Mamba-based reconstructor is the only experimentally validated direct sequence modeling approach. As detailed in overall response, MVGamba efficiently balances computational cost with long-sequence modeling capacity and offers additional valuable advantages, making it crucial for direct sequence modeling.
>
> >**Q2-1: Quantitative comparison with GS-LRM.**
>
> **A2-1:**
> + **Currently, conducting a direct and fair quantitative comparison between MVGamba and GS-LRM meets significant challenges.** We believe the performance gap is largely due to **differences in experimental settings and evaluation protocols**, such as differences in test samples, rendered views, and camera trajectories.  Notably, even the number of input views for text/image-to-3D tasks differs, as GS-LRM utilizes 6 generated images in their multi-view reconstructor.  While GS-LRM is promising, it was pre-printed on arXiv near the NeurIPS deadline without released code, leaving us insufficient time for thorough analysis or reproduction. We are eager to conduct a comprehensive comparison once GS-LRM is open-sourced or re-implemented. MVGamba will also be fully open-sourced to facilitate the 3D community.
>
> + **MVGamba's reconstruction quality can be further improved with larger model size.** Our newly trained MVGamba-B (110M) surpasses the original MVGamba-S (49M) in the manuscript by 1.9 dB in the GSO dataset for sparse-view reconstruction. We plan to further explore the potential of MVGamba with larger model size, advanced 3D representations and stronger Mamba-based architectures such as Mamba-2[2] and Samba[3].
>
> [2] Dao, T., \& Gu, A. Transformers are SSMs: Generalized models and efficient algorithms through structured state space duality. ICML2024.
>
> [3] Ren, L., et. al. Samba: Simple Hybrid State Space Models for Efficient Unlimited Context Language Modeling. arXiv:2406.07522.
>
> >**Q2-2: Is Mamba important for quality?**
>
> **A2-2: Mamba is crucial for performance improvement.** As detailed in A1-3, the Mamba-based reconstructor offers several unique advantages that enhances both quality and efficiency.
>
> To further address your concerns, Figure 3 of the rebuttal PDF shows a qualitative comparison using samples from the GRM paper. It indicates that MVGamba captures better geometries and comparable details with significantly smaller model size. Note that we could not directly compare with GS-LRM as it only presents a single-view results in Figure 7 of their paper.
>
> >**Q3: Why might GS-LRM suffer multi-view inconsistency?**
>
> **A3:** Potential multi-view inconsistency in GS-LRM may stem from:
> + **View-separated up-sampling:** GS-LRM upsamples tokens in each view separately using local deconvolution, and a large amount of Gaussians are predicted in each isolated view without 'attention' to mutual information from other views.
> + **Merge operation:** GS-LRM merges these separately upsampled Gaussians without multi-view context modeling.
>
> Note that GS-LRM was pre-printed near the NeurIPS deadline without released code. All the analysis above is based on carefully reading their paper and our empirical knowledge of this field, acknowledging potential understanding limitations without access to GS-LRM's implementation details.
>
> >**Q4: Why choose patch size 14?**
>
> **A4:** We chose a patch size of 14 based on theoretical and empirical considerations.
> + **Theoretical Insight:** MVGamba is a non-pixel-aligned model that processes causal context, requiring each patch to contain sufficient information, similar to traditional vision transformers like ViT-H-14. In contrast, GS-LRM is a pixel-aligned model and performs per-pixel Gaussian generation may require smaller patch.
>
> + **Empirical Evidence:** Ablation studies with a fixed token length of 16k on the Human-Shape subset showed patch size 14 outperformed patch size 8 by 0.71 dB PSNR. The average loss was 0.03243 for patch size 14, compared to 0.03918 for patch size 8, supporting that MVGamba benefits by patches with sufficient information.
>
> Besides, MVGamba offers additional ways to increase sequence length. Please kindly refer to the response to eshR A5.

---

> ### Author Response · Authors · 2024-08-13
>
> Dear Reviewer  XU7f,
>
> We would like to express our heartfelt gratitude for your insightful and invaluable comments, which have been of great help to us. As the discussion deadline approaches, we are eagerly looking forward to receiving your valuable feedback and comments. Ensuring that the rebuttal aligns with your suggestions is of utmost importance.
> Thank you again for your dedication to the review process and your generous support.
>
> Best regards,
>
> Authors of Submission 2178

---

> > ### Comment · Reviewer_XU7f · 2024-08-14
> >
> > Thank you for your rebuttal responses. Most of my concerns have been addressed.
> >
> > However, the Mamba-based representation did not demonstrate impressive visual quality compared to the SOTA GS-based methods, GRM and GS-LRM. Additionally, under the current sparse input setting, the input token length is not long enough to show a significant speed advantage over the GS-based methods.
> >
> > As a result, I will be maintaining my current score.

---

> ### Author Response · Authors · 2024-08-14
> **Thanks for your feedback and support!**
>
> Dear Reviewer XU7f,
>
> Thank you very much for your comments and support! It is highly encouraging to learn that our rebuttal has successfully addressed the majority of your concerns. We would like to take this opportunity to discuss two additional points with you:
>
> **Sequence Length and computational cost**. We emphasize that the Mamba-based reconstructor not only offers a speed advantage over GS-based methods but also, to our knowledge,  currently represents the only experimentally validated direct 3DGS sequence modeling approach capable of processing over 16K tokens, as discussed in our overall response. We also observe that with higher resolution inputs, there shows a substantial computational cost (heavy GPU memory) even for compressed sequence modeling, evidenced by the fact that with 512 resolution inputs, the batch size of GS LRM is 2, which is much smaller compared to 8 or 16 in regular LRM experimental settings, even with several efficient training strategy to save GPU memory, eg., gradient checkpointing, deferred backpropagation and mixed-precision training with BF16.
>
> **Visual Quality**. We believe that MVGamba could achieve comparable visual quality compared to concurrent GRM and GS-LRM with extremely smaller model size. In our future work, we intend to further enhance the reconstruction quality and explore the full potential of MVGamba through several avenues: increasing the model size, integrating more advanced multi-view diffusion models, and incorporating stronger Mamba-based architectures.
>
> We welcome any further concerns or questions you may have and are eager to engage in additional discussions or provide supplementary materials.
>
> Once again, we express our sincere gratitude for your recognition of our work and your invaluable review feedback.
>
> Best regards,
>
> Authors of Submission 2178

---

### Author Rebuttal · Authors · 2024-08-06

Dear Program Chair, Senior Area Chair, Area Chair, and Reviewers,

We sincerely appreciate the thorough review and insightful feedback provided by each reviewer. The reviewers asked perceptive questions and comments, which are answered in detail in individual responses and have improved our submission.

**In this shared response, we have attached a rebuttal PDF containing the required additional experiments. We also illustrate the two current sequence modeling paradigms for Gaussian LRMs: Compressed vs. Direct Sequence Modeling (ours) in Figure 1 of the PDF.**

Below, we elaborate on these two paradigms to highlight the unique approach of our proposed paradigm compared to recent transformer-based approaches on arXiv.

+ (a) **Compressed Sequence Modeling:** This approach tokenizes the input into a compact representation, processes it through a series of transformer blocks, and then upsamples to produce the 3DGS parameters. This paradigm is represented by the concurrent GS-LRM and GRM.

+ (b) **Direct Sequence Modeling (ours):** This approach tokenizes and expands the input into a sufficiently long sequence of tokens through cross-scan operations, which are then directly processed by a series of mamba blocks to generate the 3DGS parameters.

We believe that paradigm (b) offers several significant benefits:
+ **Efficient long sequence modeling:** It accommodates larger spatial dimensions to capture and preserve fine-grained details while mitigating information loss typically caused by upsamplers, such as the zero-padding in de-convolution layers. This benefits accurate geometry and texture reconstruction. Table 1 in the rebuttal PDF demonstrates that Mamba's linear computational complexity allows for a favorable balance between computational cost and long-sequence modeling capacity, enabling efficient processing of high-resolution inputs without prohibitive memory requirements.

+ **Easy of optimization:** It directly models the 3DGS sequence without any upsampler, hence establishing a more straightforward relationship between 3DGS parameters (e.g., position, color) and the modeled sequence. This eases the non-convex optimization in inverse graphics scenario, as discussed in pixelSplat, and helps learn accurate geometry and textures.

+ **Cross-view self-refinement:** It efficiently incorporates multi-view information causally from the initial condition onward, thereby enabling the refinement of inconsistent parts based on earlier views and generated tokens.

We would greatly appreciate it if the reviewers could review our responses. We have addressed your concerns in detail and hope our responses have answered your questions. Please let us know at your earliest convenience if you have further questions or concerns.

---

### Decision · Program_Chairs · 2024-09-25

**Decision:**

Accept (poster)

**Comment:**

Reviewers either recommend acceptance or leaning towards acceptance. All the reviewers appreciated the proposed the novel 3D generative model architecture which is memory efficient compared to existing feed-forward techniques with better reconstruction quality. Reviewers raised some concerns related to experiments several of which are addressed in the rebuttal. Reviewers did raise valid concerns and authors are encouraged to do the best of their abilities to address all the concerns in the final version.